# Evaluation of the Histomorphometric and Micromorphometric Performance of a Serum Albumin-Coated Bone Allograft Combined with A-PRF for Early and Conventional Healing Protocols after Maxillary Sinus Augmentation: A Randomized Clinical Trial

**DOI:** 10.3390/ma14071810

**Published:** 2021-04-06

**Authors:** Bálint Trimmel, Szabolcs Gyulai-Gaál, Márton Kivovics, Noémi Piroska Jákob, Csaba Hegedűs, Bence Tamás Szabó, Csaba Dobó-Nagy, György Szabó

**Affiliations:** 1Department of Oral Diagnostics, Faculty of Dentistry, Semmelweis University, Szentkirályi utca 40, 1088 Budapest, Hungary; gyulai-gaal.szabolcs@dent.semmelweis-univ.hu (S.G.-G.); szabo.bence_tamas@dent.semmelweis-univ.hu (B.T.S.); dobo-nagy.csaba@dent.semmelweis-univ.hu (C.D.-N.); 2Department of Community Dentistry, Faculty of Dentistry, Semmelweis University, Szentkirályi utca 40, 1088 Budapest, Hungary; kivovics.marton@dent.semmelweis-univ.hu; 3Department of Pathology and Experimental Cancer Research, Semmelweis University, Üllői út 26, 1085 Budapest, Hungary; noemijakob@gmail.com; 4Department of Biomaterials and Prosthetic Dentistry, Faculty of Dentistry, University of Debrecen, Egyetem tér 1, H-4032 Debrecen, Hungary; hegedus.csaba.prof@dental.unideb.hu; 5Department of Oro-Maxillofacial Surgery and Stomatology, Faculty of Dentistry, Semmelweis University, Mária utca 52, 1085 Budapest, Hungary; szabo.gyorgy@dent.semmelweis-univ.hu

**Keywords:** maxillary sinus augmentation, platelet-rich fibrin, allograft, histomorphometry, micromorphometry, resonance frequency analysis

## Abstract

The aim of this study was to compare the microarchitecture of augmented bone following maxillary sinus augmentation (MSA) after healing periods of 3 (test) and 6 (control) months using the combination of advanced platelet-rich fibrin (A-PRF) and a serum albumin-coated bone allograft (SACBA). Twenty-six patients with 30 surgical sites who required two-stage MSA were enrolled and grafted with the combination of A-PRF and SACBAs. The surgical sites were randomly allocated to the test or control group. During implant site preparation, 17 bone core biopsy samples were collected from each study group for histological, histomorphometric and micromorphometric analysis. Resonance frequency analysis was performed at the time of implant placement and 6, 8, 10, and 12 weeks postoperatively. The percentage of newly formed bone was 44.89 ± 9.49% in the test group and 39.75 ± 8.15% in the control group (*p* = 0.100). The results of the µCT analysis showed no significant differences in morphometric parameters between the study groups. The implant stability quotient was not significantly different between the two groups at 10 and 12 weeks postoperatively. Based on these findings, the total treatment time may be reduced by 3 months with the use of A-PRF and SACBAs for two-stage MSA.

## 1. Introduction

Inadequate bone quantity is a common difficulty that can compromise dental implant placement. Maxillary sinus augmentation (MSA) with the lateral window technique is a well-documented and predictable method to overcome the insufficient bone volume in the posterior maxilla caused by sinus pneumatization and alveolar ridge resorption [1,2,3]. Autologous bone (AB) was initially used as grafting material, but over the decades, different biomaterials, such as allografts, xenografts, alloplastic materials, and platelet concentrates, have also been successfully applied for this purpose [4,5,6].

For shorter healing periods, the use of AB may provide a faster remodeling ability due to its osteoinductive and osteogenic abilities; however, these properties may differ depending on the donor site [7,8]. The main disadvantage of the use of biomaterials other than AB is the lack of osteogenic properties due to their processing methods, but because of their consistent quality, the lack of donor-site morbidity bone substitute materials has proven to be a suitable alternative over the years.

The use of allografts may result in a similar bone microarchitecture as the use of AB, but their preparation technique and purification process reduce osteoinductivity, which may increase the time required for healing [9]. According to the literature, a healing period of 5–9 months is usually needed in cases of two-stage MSA with the use of allografts [6,10,11,12,13]. Serum albumin impregnation of allografts can improve osteoinductivity by restoring the albumin content of the graft that has decreased during processing [14,15,16]. Previous clinical studies have shown successful graft remodeling and observed bone microarchitecture more resembling native bone with the use of serum albumin-coated bone allografts (SACBAs) than other biomaterials [17,18,19].

The efficacy of platelet concentrates in promoting wound healing and tissue regeneration is at the centre of a recent academic debate [20,21,22]. The use of platelet-rich fibrin (PRF) was introduced by Choukroun et al. in 2006 to improve the new bone formation of allografts and reduce the healing time after MSA to 4 months [23]. In another clinical study, a composite graft of bovine xenograft and PRF was used with three different applied healing periods for two-stage MSA; the researchers reported that after 3 months of healing, it is possible to reach an appropriate primary stability of dental implants without functional loading [24]. The PRF technique is based on the use of a fibrin clot that acts as a scaffold and secures the slow release of autologous growth factors into the surrounding tissues. The lower centrifugation speed protocol (1300 rpm, 14 min) results in advanced platelet-rich fibrin (A-PRF) with increased cell content and growth factor release [25,26].

The aim of the present clinical study was to compare the bone remodeling potential of a composite graft of a SACBA and A-PRF after two-stage MSA with different healing periods. Our hypothesis was that the application of the composite graft would result in a similar bone microarchitecture and stability of implants placed in the two treatment groups with different healing periods, supporting the early implant placement protocol.

## 2. Materials and Methods

### 2.1. Study Protocol

This single-center prospective randomized controlled trial was conducted in accordance with the Helsinki Declaration and with the Consolidated Standards of Reporting Trials Statement [27]. The study protocol was reviewed and approved before patient enrollment by the Scientific and Research Ethics Committee of the Health Council of Hungary under registration number 31068-7/2018/EÜIG and by the Office of Chief Medical Officer of the National Public Health and Medical Service of Hungary under registration number 42292-5/2018/EKU and was registered in the ISRCTN registry (ISRCTN10993769). All patients participating in the study were informed about the aim of the study and the biomaterials and surgical procedures applied and signed an informed consent form.

### 2.2. Sample Size Calculation

To calculate the sample size, the G*Power 3.1 program (v.3.1.9.3, 2017, Institut für Experimentelle Psychologie, Heinrich-Heine-Universität, Düsseldorf, Germany) was used [28]. The newly formed bone percentage (NB%) was calculated as the primary outcome variable with a mean difference of 9.5% and a standard deviation of 9% between the test and control groups based on previous data [18,29]. For the expected effect size of 1.05 with an alpha level of 0.05 and a power of 80% at the 1:1 distribution ratio, 12 cases per group were used as the minimum sample size.

### 2.3. Patients

Patients were recruited from October 2018 through August 2019 at the Department of Oral Diagnostics of Semmelweis University. The following inclusion criteria were applied: systemically healthy patients, age over 18 years, need for implant-supported fixed restoration in the posterior region of the maxilla, ridge width of at least 7 mm and a residual ridge height of less than 5 mm measured on preoperative cone-beam computed tomography (CBCT). The exclusion criteria were as follows: chronic sinusitis, smoking, alcoholism, pregnancy, severe hematological disorder or disease, metabolic bone disease, dialysis, history of chronic hepatitis or liver cirrhosis, uncontrolled diabetes mellitus, bisphosphonate or immunosuppressive therapy, chemotherapy or radiotherapy.

### 2.4. Randomization

All patients underwent the same surgical procedures, the only variable between the two study groups was the applied healing time after MSA. To avoid performance bias, the sites were randomized after completing MSA by a person blinded to the intervention (Sz. Gy.) using the tossing coin method. The physicians involved in the surgical care and the patients were informed about the allocation to the test (3 months of healing) or control (6 months of healing) group at the time of suture removal.

### 2.5. A-PRF Preparation

At the beginning of the surgery, 40 mL of venous blood was drawn without anticoagulant from every patient to prepare A-PRF according to Choukroun’s technique [30]. Four tubes (A-PRF + tube, Process for PRF, Nice, France) were centrifuged for 14 min at 1300× *g* rpm (Duo Quattro Centrifuge, Process for PRF, Nice, France). After centrifugation, fibrin clots were removed from the tubes and placed in a metal box (PRF Box, Process for PRF, Nice, France) to squeeze out the liquid content and to gain membranes.

### 2.6. MSA

MSA was performed with the lateral approach according to the technique described by Vercellotti [31]. All surgeries were performed under local anesthesia (Ultracain DS Forte, Sanofi-Aventis, Paris, France). The L-shaped mucoperiosteal flap was elevated from a crestal and mesial vestibular releasing incision to access the lateral wall of the maxillary sinus. Osteotomies were carried out using a Piezoelectric surgical device (SmarThor, Megagen Co., Ltd., Daegu, South Korea), and the bone of the window was removed. The Schneiderian membrane (SM) was elevated with the use of sinus curettes. SACBA (1.5–2 cm^3^, BoneAlbumin, Orthosera Dental Zrt, Győr, Hungary) particles measuring 0.5–1.5 mm were mixed with fragments of the A-PRF membrane, and the plasma was squeezed out during membrane preparation from the metal box. Two pieces of the A-PRF membrane were placed on the SM, and the bone graft was gently packed into the created space. To cover the lateral window, the previously removed bony wall was replaced and covered with an A-PRF membrane. The mucoperiosteal flap was mobilized to allow tension-free closure with single interrupted nonresorbable sutures (Dafilon 4/0 DS19, B. Braun AG, Melsungen, Germany). All augmentation procedures were performed by the same experienced surgeon (B.T.). Antibiotics (1 g amoxicillin-clavulanic acid twice a day for 7 days), anti-inflammatory drugs (275 mg naproxen 3 times a day for 3 days) and chlorhexidine mouthwash (twice a day for 7 days) were prescribed. Sutures were removed 7 days after MSA. During the healing period, the surgical areas were not loaded with any type of prosthesis. After 3 (test group) or 6 months (control group) of healing CBCT scans (Planmeca ProMax 3D CBCT, Planmeca Oy, Helsinki, Finland) were performed from all surgical sites prior to implant placement. Figure 1 shows an intervention and the pre- and postoperative CBCT images of the surgical area from the test group.

### 2.7. Implant Placement

All interventions were performed by the same surgeon (B.T.) under local anesthesia (Ultracain DS Forte, Sanofi-Aventis, Paris, France). A midcrestal incision was made to raise a full-thickness flap. A modular trephine drill (Full-Tech Kft, Szigetszentmiklós, Hungary) designed for the study (internal 2 mm, outer 2.7 mm in diameter and 10 mm in length) was used as an initial drill to collect bone core biopsy samples, as shown in Figure 2. The implant site preparation was continued with the manufacturer’s drills according to their protocol. Directly after implant placement (Straumann SP RN implants with Ti-SLA surface, Straumann GmBH, Basel, Switzerland), resonance frequency analysis (RFA) was carried out to measure implant stability. A one-stage healing protocol was applied, and interrupted nonresorbable sutures were used to close the flap around the gingiva formers. Antibiotics (1 g amoxicillin-clavulanic acid twice a day for 7 days), anti-inflammatory drugs (275 mg naproxen 3 times a day for 3 days) and chlorhexidine mouthwash (twice a day for 7 days) were prescribed, and the sutures were removed 7 days after implant placement.

### 2.8. RFA

The implant stability quotient (ISQ) was measured immediately after implant placement and postoperatively at 6, 8, 10, and 12 weeks with SmartPegs and an RFA device (Osstell IDx, Osstell AB, Göteborg, Sweden). The measurements were performed in the buccolingual and mesiodistal directions according to the manufacturer’s protocol, and the lower values were saved in the RFA device.

### 2.9. Micromorphometric Analysis

After biopsy removal, the bone core biopsy samples were placed in a 0.3 mL microcentrifuge tube (Eppendorf tube, Merck KGaA, Darmstadt, Germany) and fixed in 10% formaldehyde, 0.1 M phosphate buffer saline (PBS), pH 7.3 solution. The tubes were code-masked to facilitate blind histomorphometric and micromorphometric analysis. The bone samples were scanned using a microcomputed tomography (μCT) scanner (Bruker 1272 X-ray microtomograph, Bruker µCT, Kontich, Belgium). Scanning was carried out at a resolution of 5.9 µm (60 kV, 166 µA). For image noise reduction, an Al 0.25 mm aluminum filter was used. The average scan duration was 30 min. Raw images were reconstructed by using NRecon software (v.1.7.4.6., Bruker µCT, Kontich, Belgium). The ring artifact correction was 13, and the beam-hardening correction was 25%. CTAn software (v.1.17.7.2, Bruker µCT, Kontich, Belgium) was used to perform 3D morphometric analysis. The definitions of the relevant morphometric variables for the study are listed in Table 1 [32,33]. The quantitative analysis was performed by two blinded examiners (B.T.Sz., Cs. D-N). For each sample, the complex 3D structure was analyzed to identify the pristine bone and the augmented area. The transitional zone (80–120 segment) between the host and the augmented area was identified on these reconstructed images and was excluded from the quantitative analysis. The regions of interest (ROIs) of quantitative analysis were the portion of augmented area compared to the portion of pristine residual ridge. The volume of ROIs differed among the bone core biopsy samples, which was influenced by the residual ridge height of the surgical sites and the length of the biopsy; therefore, volume-independent metrics were used for the analysis.

### 2.10. Histology and Histomorphometric Analysis

Following µCT scanning, the bone core biopsy samples were histologically processed. The samples were embedded in paraffin after decalcination and dehydration, and 20 µm sections were prepared. The sections were stained with hematoxylin-eosin stain and digitalized by a slide scanner (Panoramic 1000, 3DHISTECH Ltd., Budapest, Hungary) for histological evaluation. The digital images were transferred to CaseViewer 2.4 (3DHISTECH Ltd., Budapest, Hungary) for histomorphometric analysis by a blinded examiner. The images were evaluated at 150× magnification to identify the margin between pristine bone and the augmented area. The area of pristine bone was excluded from the analysis. Two representative slides of each histologic sample were selected from the augmented areas. The percentages of newly formed bone (NB), residual graft particles (RG) and nonmineralized tissue (NMT) were determined based on manual segmentation with Adobe PhotoShop (Adobe System Inc., San Jose, CA, USA) and ImageJ for Windows (ImageJ 1.45, 2011, Wayne Rasband, US National Institute of Health, Bethesda, MD, USA) software.

### 2.11. Statistical Analysis

Statistical analysis was performed using IBM SPSS Statistics 25 software (IBM Corporation, New York, NY, USA). All data are expressed as the mean ± standard deviation. The Shapiro–Wilk test was used to assess the normality of the data distribution. The histomorphometric variables showed normal distribution; therefore, one-way ANOVA was used for statistical analysis. The ISQ values showed a normal distribution at the time of implant placement and were analyzed by one-way ANOVA. Postoperatively, at 6, 8, 10, and 12 weeks of evaluation, the ISQ values showed a non-normal distribution, and these data were analyzed by the independent-samples Mann–Whitney U test. Multiple comparison tests were used to analyze the four datasets of µCT data. The values of bone volume fraction (BV/TV), bone surface/volume ratio (BS/BV), trabecular separation (Tb.Sp), total porosity (Po(tot)) and open porosity (Po(op)) showed normal distributions and were analyzed by one-way ANOVA with Bonferroni and Tukey HSD post hoc tests. The values of the bone surface/volume ratio (BS/TV), trabecular thickness (Tb.Th), Trabecular bone pattern factor (Tb.Th) and connectivity (Conn) showed a nonnormal distribution and were analyzed by the independent-samples Kruskal–Wallis test with Bonferroni correction for multiple tests. Differences were considered statistically significant at *p* < 0.05.

## 3. Results

### 3.1. Patient Characteristics

Twenty-six patients with 30 MSA were enrolled in this study, as shown in Figure 3. The patient demographics, interventions, sites of implant placement and bone core biopsy harvesting are summarized in Table 2.

The mean age was 57.93 ± 7.79 years in the test group and 55.33 ± 8.55 years in the control group. The residual ridge height was measured on CBCT sagittal sections between the alveolar crest and the lowest point of the maxillary sinus. For determination of the maxillary sinus width on CBCT coronal section, a horizontal line was drawn through the line of the palatonasal recess, and at this height, the mesiodistal width of the maxillary sinus at the site of the first upper molar was measured. The characteristics of the maxillary sinuses were similar in the two groups; the residual ridge height was 2.93 ± 1.14 mm in the test group and 3.48 ± 1.04 mm in the control group, and the sinus width was 15.06 ± 0.85 mm in the test group and 14.57 ± 1.41 mm in the control group.

### 3.2. MSA

In three cases (two in the test group and one in the control group) of the 30 MSAs, a small perforation (diameter less than 5 mm) of the SM was observed. Using A-PRF membranes, the SM perforations were covered, and the augmentation procedures were completed without postoperative complications.

### 3.3. Implant Placement

During the second surgery, 17–17 bone core biopsy samples were collected from the implant placement sites in both groups, and a total of 53 dental implants were placed: 26 in the test group and 27 in the control group. Two dental implants from the control group were lost during the healing period (one at 6 weeks and one at 8 weeks postoperatively). In these two cases, new implants were placed 3 months later with successful osseointegration. In all cases, screw-retained crowns or bridges were made. After 12 months of loading, the overall implant survival rate was 96.2%.

### 3.4. RFA

ISQ was measured immediately after implant placement and postoperatively at 6, 8, 10, and 12 weeks. The implants showed increasing ISQ values during the healing period with the exception of the two lost implants. Significantly higher values were recorded in the control group than in the test group at 6 weeks (74.22 ± 4.52 and 68.52 ± 7.35) and 8 weeks (75.70 ± 4.76 and 72.00 ± 7.16) postoperatively. At the time of implant placement (68.92 ± 7.56 and 72.20 ± 6.30) and after 10 (74.26 ± 5.79 and 75.74 ± 4.93) and 12 (75.96 ± 4.75 and 76.96 ± 4.31) weeks of healing, ISQ values did not show a statistically significant difference between the test and control groups. More details are shown in Table 3.

### 3.5. Micromorphometry Analysis

Within bone core biopsy samples, μCT reconstruction showed a marked difference in the appearance of the pristine maxillary bone and the augmented areas in both groups. The images of pristine bone from both study groups showed thick, continuous trabeculae with wide medullary spaces. The radiolucency of SACBA particles and the NB of the augmented areas were not separable; however, the complex 3D structure was based on a network of thinner trabeculae and abundant nodes. The pristine bone and augmented parts of the samples were differentiated based on these characteristics and were compared with each other. The transition zone (80–120 segment) between the native and augmented bone was identified and excluded from the quantitative analysis for more accurate evaluation.

Morphometric data of the pristine maxillary bone and the augmented areas of both study groups were compared. The analysis showed no statistically significant difference in comparing the grafted areas of the test and control groups, and no statistically significant difference was observed in either of the morphometric parameters between the pristine bone area of the two groups. The BS/BV, BS/TV and Conn values were significantly higher, and the Tb.Th was significantly thinner in the augmented areas, as shown in Figure 4. Similar values were observed between the native and augmented bone for the other investigated micromorphometric parameters. Detailed results of the micromorphometric analysis are shown in Appendix A, Table A1.

### 3.6. Histological and Histomorphometric Analysis

A total of 34 bone core biopsy samples were obtained from 30 MSAs, and both groups represented 17–17 specimens. Based on histological analysis, signs of gradual graft resorption and remodeling were observed in the augmented areas, which occurred in both groups without foreign body reaction or inflammation. RG particles were surrounded by NB and NMT. NB was 44.89 ± 9.49% and 39.75 ± 8.15%, RG was 12.52 ± 6.25% and 15.67 ± 6.92%, and NMT was 42.59 ± 12.48% and 44.58 ± 13.35% for the test and control groups, respectively, with no statistically significant differences in either of the tissue labels between groups. A representative histological section and the histomorphometric data are presented in Figure 5.

## 4. Discussion

This randomized controlled clinical trial was conducted to investigate the bone remodeling potential of a composite graft of a SACBA and A-PRF after MSA with different healing periods. Micromorphometric, histomorphometric and RFA analyses were used to evaluate the characteristics of augmented areas and the changes in implant stability during the osseointegration process. The results of the present study suggested that the combination of a SACBA and A-PRF constitutes a suitable biomaterial for MSA, which allows implant placement after 3 months with the same results as the conventionally used 6-month healing period. In both groups after 8 weeks of implant placement, the ISQ values reached 70, indicating adequate osseointegration for the onset of prosthetic workflows.

MSA is one of the most commonly performed regenerative treatments in oral surgery and was originally described with the use of AB as a graft material [2,34]. The main reasons for the development of biomaterials to replace AB are donor-site morbidity and limited availability; however, the osteogenic capabilities of AB seem to be unique and hardly substitutable [35]. There is extensive literature on different types of biomaterials used for MSA and comparisons with each other or AB in randomized clinical trials or in systematic reviews [36,37]. The NB formation capacity of biomaterials can be improved by the addition of autologous cell concentrates or AB, providing similar histomorphometric results as the use of AB alone if a 5–8 month healing period is applied after MSA [38]. The available biomaterials provide a suitable alternative in terms of new bone-forming ability, long-term tissue and volume stability, or dental implant survival rate [4,5,6].

The time of bone graft remodeling is influenced by the characteristics of the biomaterials used [36]. Several recently published clinical trials have focused on the optimal healing timing for implant placement [24,29,39]. Butz et al. reported that implant placement after 2 months of healing is possible if a combination of bovine xenografts and synthetic peptides in a sodium hyaluronate carrier is used for two-stage MSA [29]. Tatullo et al. used bovine xenografts and PRF for two-stage MSA and reported that good primary stability of endosseous implants without functional loading is achievable after 3 months of healing [24]. Both randomized clinical trials found similar NB formation with the applied early and conventional healing protocols.

Although allograft-processing methods maintain the original microstructure of the bone and a significant portion of the apatite content, they destroy the cell content and eliminate some of the osteoinductive proteins to reduce immunogenic activity, leading to slower remodeling abilities and increased graft resorption [9]. For this reason, a healing period of 5 to 9 months is most commonly used for two-stage MSA with allografts before dental implant placement [10,11,12,13]. To overcome the reduction in osteoinductive properties and slower remodeling abilities, human serum albumin impregnation of freeze-dried bone allografts was investigated in vitro and in vivo [14,15,16]. Bone marrow-derived mesenchymal stem cells require a 10% concentration of albumin in the cell culture medium for adhesion and differentiation [40]. It has been proven that albumin is necessary for early bone healing processes and is physiologically present in native bone [40]. Previous studies with SACBAs in the field of orthopedics and oral surgery have reported uncomplicated healing with a high degree of graft remodeling [17,19,41].

A total of eight previous studies examined the benefits of using PRF for the improvement of new bone formation in two-stage MSA. Five of them found no histomorphometrical differences between bone substitutes and composite grafts containing PRF and bone substitutes [24,42,43,44,45]. Two studies reported similar histomorphometric results in the control and PRF-containing test groups, but shorter healing times were applied in both of the test groups [23,46]. A recent publication reported significantly higher NB values using a composite graft of deproteinized bovine bone material and PRF with a healing time of 4 months after MSA [39]. However, it is important to note that each study has applied different PRF production protocols regarding the centrifugation speed, force and time. The most important details of these studies are summarized in Appendix A, Table A2.

The use of PRF may provide an opportunity to improve the osteoinductive abilities lost in the manufacture of bone substitutes by releasing autologous growth factors into the surrounding tissues, thereby promoting faster remodeling [23,39,47]. The A-PRF obtained as a result of the low-speed centrifugation protocol increases the total amount of growth factors embedded in the scaffold and prolongs their release time by 14 days, which can be advantageous in guided bone regeneration [26,48].

In the present study, the use of a composite graft of a SACBA and A-PRF was investigated for MSA with early and normal healing times to evaluate the microarchitecture of the augmented areas.

The results of histomorphometric analysis showed similar values of NB, RG and NMT in both groups. The achievable high percentage of NB (44.89 ± 9.49% and 39.75 ± 8.15%; *p* = 0.100, for the test and control groups) allows re-entry and dental implant placement to be performed 3 months after MSA [11]. Compared to the literature data, the low percentage of RG (12.52 ± 6.25% after 3 months and 15.67 ± 6.92% after 6 months) may indicate faster remodeling and may be a possible effect of A-PRF [13,23]. Complete remodeling of a SACBA alone may require 6–12 months according to the literature [17,18].

The µCT analysis data were consistent with the histomorphometric analysis results. Morphometric data from pristine and augmented bone were compared at both applied healing periods. According to the analysis, the grafted areas of the test and control groups showed similar microarchitectures. The SACBA- and A-PRF-augmented bone consisted of progressively resorbed RG particles surrounded by thin, lamellar and cylindrical NB trabeculae with an extensive bone marrow network. In the µCT reconstruction of bone core biopsy samples, the residual bone of the alveolar ridge and the augmented bone was distinguished by the differences in their microarchitecture. Pristine bone is characterized by a contiguous network of mature bone trabeculae, whereas the augmented bone is characterized by the convoluted network of thin, immature bone trabeculae, and isolated bone-like formations (partially resorbed RG particles).

Gradual absorption and transformation of biomaterials into NB is accompanied by an increase in bone surface. The BS/BV and BS/TV values were significantly higher in the augmented bone, probably due to the extensive remodeling of RG particles. The Tb.Th was significantly thinner in the augmented areas, which may suggest that a longer time is required for graft maturation to reach these parameters, namely, the values of the pristine bone. The significantly higher value of Conn in the grafted areas also suggests a compacted, not completely remodeled microstructure maintained by RG particles. For the parameters of BV/TV, Tb.Sp, Tb.Pf, Po(tot) and Po(op), similar values were observed between pristine and augmented bone, suggesting that the addition of A-PRF to SACBAs may have facilitated graft remodeling compared to the data of a previous µCT study [18].

Based on previous literature data, early loading protocol for moderately rough-surfaced dental implants (4–12 weeks healing after implant placement) can be successfully applied in the areas of posterior maxilla, although the healing time of individual implants can be affected by several factors [49]. From a clinical point of view, investigating implant stability during the healing process can be useful to confirm the adequate healing time for implant placement (primary stability) or for prosthetic loading (secondary stability) after two-stage MSA. ISQ values provide information on implant fixation, and the measurement can be affected by the degree of bone-implant contact, osseointegration, and bone quality [50,51,52]. The results of the present study showed similar values between the two groups immediately after implant placement and at 10 and 12 weeks after implant placement as well as statistically higher ISQ values in the control group at 6 and 8 weeks of healing. These differences are probably related to the physical parameters of the NB, which explains the higher values associated with a longer healing time [53,54]. The decline in ISQ values at week 6 in the test group can be explained by bone remodeling processes that cause a decrease in the primary stability achieved during insertion. The rising ISQ values observed in the subsequent weeks were signs of secondary stability and osseointegration achieved by NB formation on the implant surface [55,56]. The lack of measurement of two implants lost in the control group and therefore not included in the statistics may be the reason no similar decrease in the ISQ values was observed in the control group. Eight weeks after implant placement, the mean values of both groups exceeded the ISQ value of 70, providing adequate stability for the onset of prosthetic workflows [53].

These results show that the use of a composite graft of A-PRF and a SACBA for MSA improves graft remodeling kinetics, and the augmented bone microarchitecture does not differ significantly between 3- and 6-month healing periods, allowing earlier implant placement as a reliable alternative to the conventional treatment protocol.

## 5. Conclusions

The results of the present study suggest that a composite graft of a SACBA and A-PRF is a suitable biomaterial for MSA and allows earlier implant placement based on the similar histomorphometric and micromorphometric parameters of the augmented areas with 3 or 6 months of healing. According to these findings, the total treatment time of two-stage MSA may be reduced by 3 months, however further studies may be needed to confirm these results.

## Figures and Tables

**Figure 1 materials-14-01810-f001:**
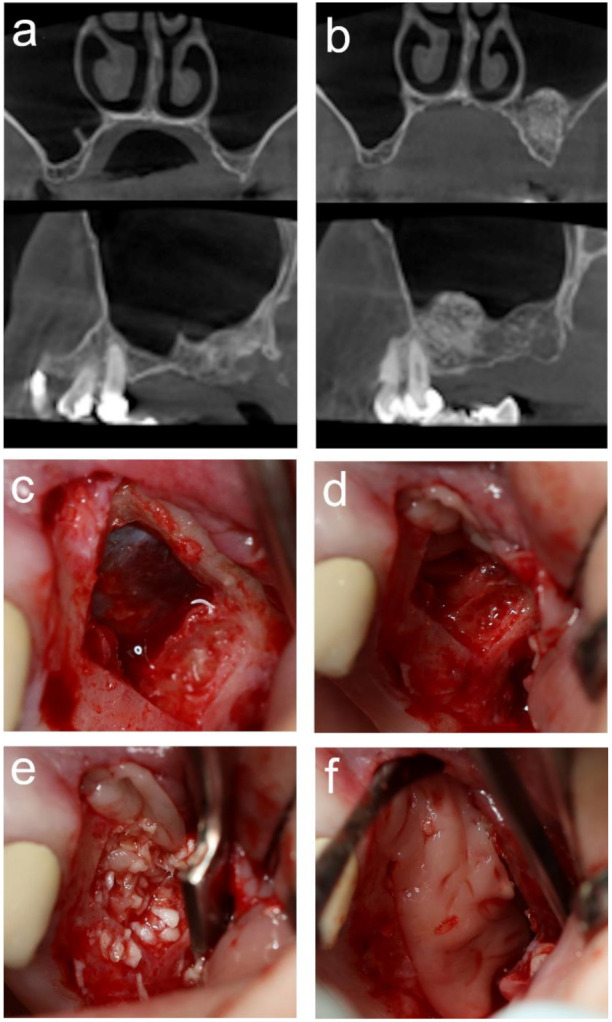
Representative cone-beam computed tomography (CBCT) images are shown from the test group before maxillary sinus augmentation. (MSA) (**a**) and after 3 months of healing (**b**). The main steps of MSA were as follows: piezoelectric osteotomy was carried out, and the bony window was removed to expose the Schneiderian membrane (SM) (**c**). SM was elevated and then covered with two pieces of advanced platelet-rich fibrin (A-PRF) membrane (**d**). Serum albumin-coated bone allograft particles were mixed with A-PRF and gently packed into the created space (**e**). The lateral window was covered with the previously removed bony wall and an A-PRF membrane (**f**).

**Figure 2 materials-14-01810-f002:**
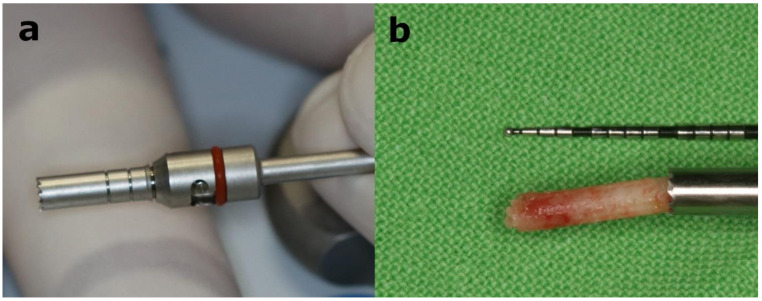
A 10 mm working length modular trephine drill with 2 mm internal and 2.7 mm outer diameter (**a**) was used as initial drill for implant placement to collect bone core biopsy samples for histological, histomorphological and micromorphological analysis (**b**).

**Figure 3 materials-14-01810-f003:**
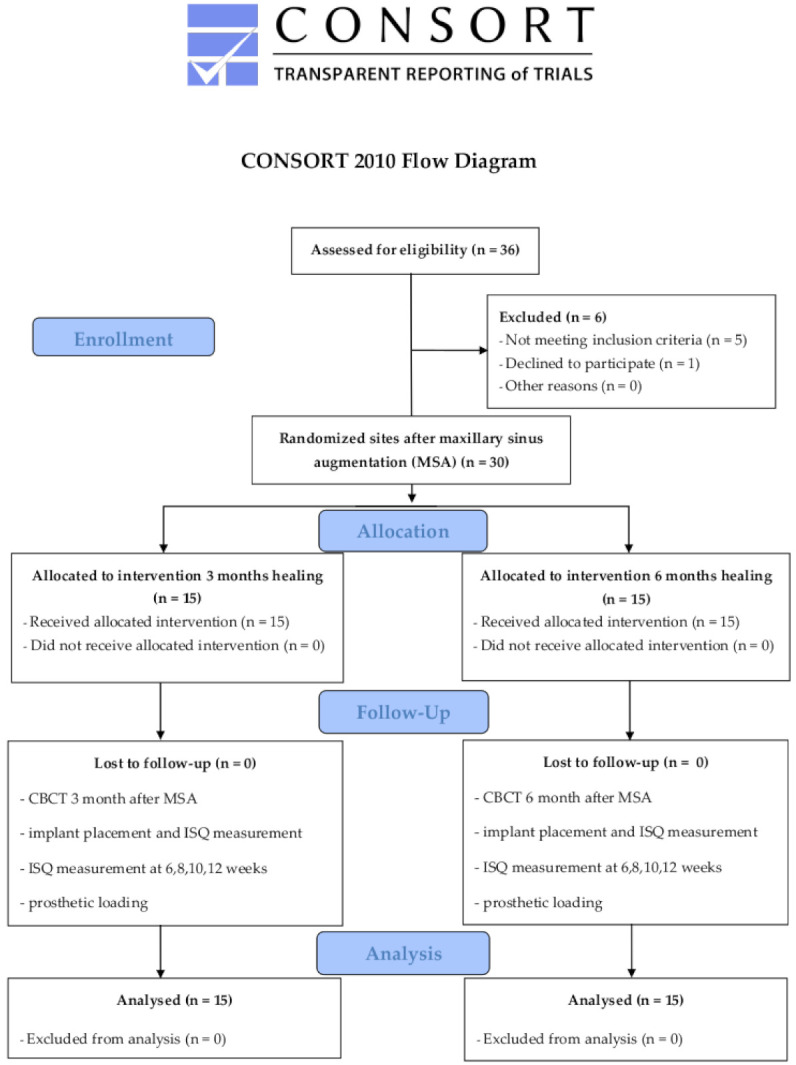
CONSORT flow diagram.

**Figure 4 materials-14-01810-f004:**
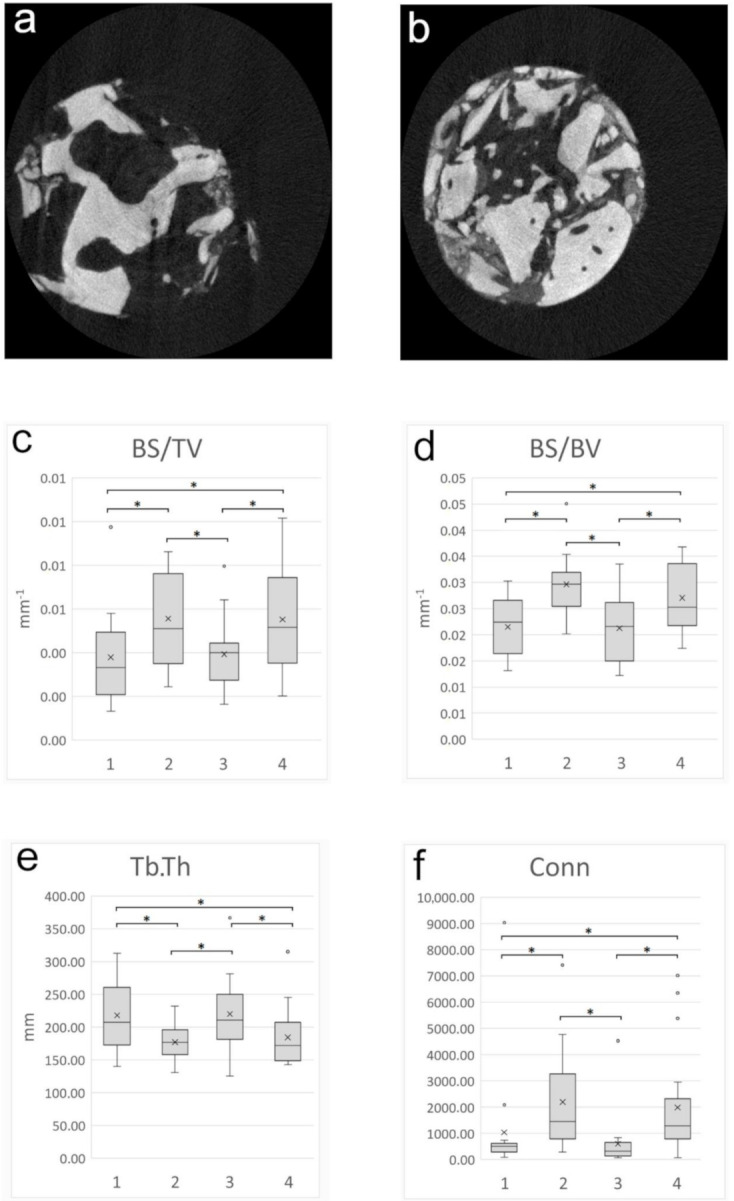
Representative µCT images of the bone core biopsy samples are shown at the level of the pristine maxillary bone (**a**) and the level of the augmented bone (**b**) after 3 months of healing. Panels (**c**–**f**) show significant comparisons between the pristine maxillary bone and augmented bone based on micromorphometric data. Statistical significance (*p* < 0.05) is marked by an asterisk. The following abbreviations were used: pristine bone in the 3-month healing group (1), augmented areas in the 3-month healing group (2), pristine bone in the 6-month healing group (3), augmented areas in the 6-month healing group (4), bone surface/volume ratio (BS/BV), bone surface density (BS/TV), trabecular thickness (Tb.Th) and the connectivity (Conn).

**Figure 5 materials-14-01810-f005:**
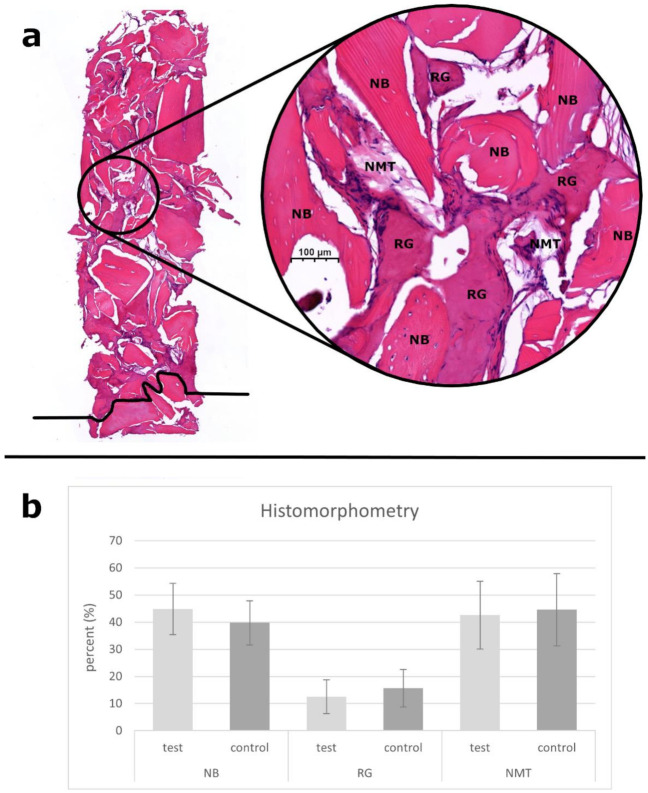
A representative histological section (**a**) with hematoxylin-eosin staining of a bone core biopsy sample after 3 months of healing (test). The black line indicates the level of the pristine maxillary bone, which was excluded from the analysis. Panel (**b**) shows the histomorphometry data (mean ± standard deviation) of the stained sections. Abbreviations: newly formed bone (NB), residual graft particles (RG), nonmineralized tissue (NMT).

**Table 1 materials-14-01810-t001:** The characteristics of morphometric variables investigated in the present study and calculated by CTAn. software (according to the manual Bruker MicroCT Morphometric parameters measured by CT-analyzer software 1.15.4.0 by Bruker microCT).

Abbreviation	Variable	Description	Standard Unit
BV/TV	Bone volume fraction	Relative volume of calcified tissue in the selected volume of interest (VOI).	%
BS/TV	Bone surface density	The ratio of surface area to total volume measured in 3D, within the VOI.	mm^−1^
BS/BV	Bone surface / volume ratio	Surface to volume ratio of calcified tissue or “specific surface” is useful for characterising the complexity and thickness of structures.	mm^−1^
Tb.Th	Trabecular thickness	Mean thickness of trabeculae, assessed using direct 3D methods.	mm
Tb.Sp	Trabecular separation	Trabecular separation is essentially the thickness of the spaces as defined by binarisation within the VOI.	mm
Tb.Pf	Trabecular bone pattern factor	This is an index of connectivity of trabecular bone; it calculates an index of relative convexity or concavity of the total bone surface, on the principle that concavity indicates connectivity (and the presence of “nodes”), and convexity indicates isolated disconnected structures (struts).	1/mm
Po(tot)	Total porosity	Total porosity is the volume of all open plus closed pores as a percent of the total VOI volume.	%
Po(op)	Open porosity	Percent open porosity is the volume of open pores as a percent of the total VOI volume.	%
Conn.	Connectivity	One useful and fast algorithm for calculating the Euler connectivity in 3D is the “Conneulor”. It measures what might be called “redundant connectivity”, the degree to which parts of the object are multiply connected. It is a measure of how many connections in a structure can be severed before the structure falls into two separate pieces.	none

**Table 2 materials-14-01810-t002:** Demographics and clinical data of the patients. Abbreviations: male (M), female (F), right (R), left (L).

Patient	Age (years)	Gender	Maxillary Sinus	Residual Ridge Height	Maxillary Sinus Width	Healing Protocol	Implants Position (FDI)	Position of Bone Core Biopsy (FDI)
1	41	F	R	4.7 mm	13.9 mm	6 months	14, 16	16
2	43	F	L	4.2 mm	14.8 mm	6 months	25, 26	26
3	54	F	R	4.8 mm	13.8 mm	6 months	14, 16	16
			L	4.5 mm	13.4 mm	3 months	24, 26	26
4	63	F	R	2.2 mm	14.1 mm	3 months	15, 16	16
5	56	M	L	4.6 mm	16.2 mm	3 months	26	26
6	46	F	R	1.9 mm	15.6 mm	6 months	16	16
7	60	F	L	3.8 mm	12.7 mm	6 months	25, 26	26
8	51	F	R	2.3 mm	14.7 mm	3 months	15, 16	15, 16
9	62	F	L	3.7 mm	14.8 mm	6 months	25, 26	26
10	67	M	R	4.5 mm	13.8 mm	6 months	17	17
11	63	M	L	1.6 mm	16.5 mm	3 months	24, 26	26
12	60	F	R	3.3 mm	11.8 mm	6 months	16	16
13	68	F	R	1.9 mm	15.3 mm	3 months	14, 16	16
14	60	M	L	4.2 mm	15.2 mm	6 months	26, 27	26, 27
15	56	F	L	3.9 mm	14.5 mm	3 months	26	26
16	63	F	R	1.9 mm	14.6 mm	6 months	15, 16	16
			L	2.5 mm	14.7 mm	3 months	24, 26	26
17	61	M	L	1.2 mm	15.2 mm	3 months	26	26
18	46	M	R	3.1 mm	14.9 mm	3 months	14, 17	17
			L	3.4 mm	15.2 mm	6 months	24, 26	26
19	41	F	L	4.2 mm	15.7 mm	3 months	26, 27	26
20	68	F	L	4.1 mm	15.5 mm	6 months	15, 17	17
			R	3.3 mm	14.6 mm	3 months	24, 27	27
21	53	F	R	1.7 mm	16.9 mm	6 months	15, 16	16
22	54	M	R	2.4 mm	15.9 mm	3 months	15, 16	16
23	61	M	L	1.9 mm	15.8 mm	3 months	25, 26	25, 26
24	56	M	R	2.4 mm	16.8 mm	6 months	15, 16	16
25	64	F	R	4.3 mm	14.4 mm	3 months	16	16
26	51	F	L	3.6 mm	13.2 mm	6 months	26, 27	26, 27

**Table 3 materials-14-01810-t003:** The results of implant stability quotient (ISQ) measurement at different time points after implant placement. Statistical significance (*p* < 0.05) is highlighted in bold. Abbreviations: one-way ANOVA (#), independent-samples Mann–Whitney U test (##).

	Group	*N*	Mean	Std. Deviation	*p*-Value
**ISQ week 0**	test	24	68.92	7.56	0.105 ^#^
	control	25	72.20	6.30	
**ISQ week 6**	test	25	68.52	7.35	**0.003** ^**##**^
	control	24	74.22	4.52	
**ISQ week 8**	test	26	72.00	7.16	**0.041** ^**##**^
	control	25	75.70	4.76	
**ISQ week 10**	test	26	74.26	5.79	0.501 ^##^
	control	24	75.74	4.93	
**ISQ week 12**	test	26	75.96	4.75	0.345 ^##^
	control	24	76.96	4.31	

## Data Availability

Data is contained within the article.

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
