# Peer review of "Evaluation of the Histomorphometric and Micromorphometric Performance of a Serum Albumin-Coated Bone Allograft Combined with A-PRF for Early and Conventional Healing Protocols after Maxillary Sinus Augmentation: A Randomized Clinical Trial"

_materials, 2021, doi:10.3390/ma14071810_

Round 1

Reviewer 1 Report

The study of “Evaluation of the histomorphometric and micromorphometric performance of a serum albumin-coated bone allograft com-bined with A-PRF for early and conventional healing protocols after maxillary sinus augmentation: a randomized clinical trial” used micro-CT, histomorphometric analysis to compare the microarchitecture of augmented bone following maxillary sinus augmentation (MSA) after 3 (test) and 6 (control) months using the combination of advanced platelet-rich fibrin (A-PRF) and a serum albumin-coated bone allograft (SACBA). Additionally, this study also used osstell instrument to evaluate the implant stability quotient (ISQ) after implant placement at 6, 8, 10, and 12 weeks.

Overall, it is an very clinical orientated and applied study. The methodology is sound. The results is reasonable. The conclusion is partially drawn from the results. I have only some minor concerns about this study.

  1. Osstell instrument is a reliable tool for detecting osseointegration status between implant and bone by measuring implant stability. However, in this study the longest duration is set at 3 months (12 weeks). Usually, the healing process of dental implant is about 4-6 months. Why was the original experimental design set for 3 months?
  2. In Micro-CT analysis, how to distinguish the structure between pristine bone and grafted bone? How about the threshold of gray scale value between pristine bone and grafted bone?
  3. In the discussion section (page 14), The meaning of “extensive remodeling of RG particles” is not clear. Please express it more clearly for how it influence the values of BS/BV and BS/TV.
  4. The content of the 6. Patent is blank
  5. The format of the names of authors in the reference # 48 is not correct.

Reviewer 2 Report

The manuscript submitted to Materials entitled “Evaluation of the histomorphometric and micromorphometric performance of a serum albumin-coated bone allograft combined with A-PRF for early and conventional healing protocols after maxillary sinus augmentation: a randomized clinical trial” is an original article which aim to  compare the microarchitecture of augmented bone following maxillary sinus augmentation (MSA) after healing periods of 3 (test) and 6 (control) months using the combination of platelet-rich fibrin (PRF) and a serum albumin-coated bone allograft (SACBA).

On my opinion the article is interesting and well written. However, I highlighted some issues.

  • English language: Minor spell check is required.
  • Abstract: Please structure the abstract to attract the reader's attention,and to adapt it accordingly
  • Introduction: This section needs improvement. What are autologous platelet concentrates? Is there evidence on their effects on tissue regeneration? I would suggest inserting the following sentence and reference on page 2: <<Efficacy of platelet concentrates in promoting wound healing and tissue regeneration is at the centre of a recent academic debate [doi: 1007/s00784-020-03702-w].>>.
  • Materials and Methods: There are no subsections on the randomization of patients and on the different interventions performed to compare! How were they randomized? What is the difference between the two groups? Explain it in detail.
  • Results: To be modified in accordance with the section "Materials and methods".
  • Discussion: The article is original even if there are several studies in the literature that evaluate the effect of PRF and other platelet concentrate in MSA and alveolar ridge augmentation. It would be interesting to insert a table in the discussion section in which the authors list the various studies on the subject and the objectives and results reported.
  • Conclusion: The conclusions consistent with the evidence and arguments presented but further studies will be needed to confirm the authors' hypotheses.
  • Summary of abbreviations required at the end of the manuscript prior to “Reference” section.
  • Figures and Tables: Please improve figures resolution. Consort flow diagram is not visible in its entirety.

After making the indicated changes, I am available for a second round of peer review.

Round 2

Reviewer 1 Report

The authors have answered all the concerns I have. I have no more questions about it. 

Reviewer 2 Report

After the changes made, the article is suitable for publication.